# Transcriptome Analysis Reveals Key Seed-Development Genes in Common Buckwheat (*Fagopyrum esculentum*)

**DOI:** 10.3390/ijms20174303

**Published:** 2019-09-03

**Authors:** Hongyou Li, Qiuyu Lv, Jiao Deng, Juan Huang, Fang Cai, Chenggang Liang, Qijiao Chen, Yan Wang, Liwei Zhu, Xiaona Zhang, Qingfu Chen

**Affiliations:** 1Research Center of Buckwheat Industry Technology, Guizhou Normal University, Guiyang 550001, China; 2School of Big Data and Computer Science, Guizhou Normal University, Guiyang 550025, China

**Keywords:** common buckwheat, seed development, seed size, RNA-Seq, DEGs, transcription factors

## Abstract

Seed development is an essential and complex process, which is involved in seed size change and various nutrients accumulation, and determines crop yield and quality. Common buckwheat (*Fagopyrum esculentum* Moench) is a widely cultivated minor crop with excellent economic and nutritional value in temperate zones. However, little is known about the molecular mechanisms of seed development in common buckwheat (*Fagopyrum esculentum*). In this study, we performed RNA-Seq to investigate the transcriptional dynamics and identify the key genes involved in common buckwheat seed development at three different developmental stages. A total of 4619 differentially expressed genes (DEGs) were identified. Based on the results of Gene Ontology (GO) and KEGG analysis of DEGs, many key genes involved in the seed development, including the Ca^2+^ signal transduction pathway, the hormone signal transduction pathways, transcription factors (TFs), and starch biosynthesis-related genes, were identified. More importantly, 18 DEGs were identified as the key candidate genes for seed size through homologous query using the known seed size-related genes from different seed plants. Furthermore, 15 DEGs from these identified as the key genes of seed development were selected to confirm the validity of the data by using quantitative real-time PCR (qRT-PCR), and the results show high consistency with the RNA-Seq results. Taken together, our results revealed the underlying molecular mechanisms of common buckwheat seed development and could provide valuable information for further studies, especially for common buckwheat seed improvement.

## 1. Introduction

Seed or grain development is one of the most crucial processes in plant development. It not only determines the successful racial continuation of seed plants, but also affects the final seed yield and seed quality [1,2]. Generally, seed development can be broadly divided into two phases: Embryogenesis and maturation [3,4]. The embryogenesis phase mainly involves cell division and expansion, while maturation is usually an accumulation process of seed storage materials [3]. Seed development is predominantly regulated by genetic factors and is also greatly influenced by physiological pathways and environmental cues. At the physiological level, phytohormones—including auxins, cytokinins (CKs), gibberellins (GAs), brassinolides (BRs), abscisic acid (ABA), and ethylene (ET)—contribute to seed development [2]. At the molecular level, in the past two decades, many genes involved in seed development have been identified and their molecular pathways deciphered in the model plants *Arabidopsis* and rice through analysis of mutants, transcriptomes, and quantitative trait locus (QTLs) [2,5,6,7,8,9]. Furthermore, in the last few years, investigations of the molecular mechanisms of seed development have also been undertaken in some crop plants such as maize [10,11], wheat [12,13], soybean [14,15,16], barley [4], Tartary buckwheat [17,18], chickpea [19], and *Brassica napus* [20,21] by transcriptome analysis. These studies have identified large numbers of related regulatory and functional genes, and provided a better understanding of the molecular mechanisms of seed development. 

Common buckwheat (*Fagopyrum esculentum*) is an annual minor crop which belongs to the eudicot family Polygonaceae, mainly cultivated in temperate regions of Asia, Europe, and North America [22]. As a major part of the human diet, common buckwheat seeds contain high levels of starch, protein, dietary fibers, vitamins, minerals, and flavonoids, which are beneficial to human health [23,24,25]. Furthermore, the seed of common buckwheat is larger than that of another cultivated buckwheat, tartary buckwheat (*Fagopyrum tataricum*). Therefore, it is of considerable interest to identify genes and dissect the molecular mechanisms of seed development in common buckwheat, as this will contribute to common buckwheat and tartary buckwheat seed improvement, including seed formation, morphogenesis, size, and nutrient accumulation. To date, several studies have investigated the transcriptional profile of the developing seeds of common buckwheat though RNA-Seq [26,27,28]. However, these studies only focused on one developmental time point of common buckwheat seed, and the differentially expressed genes that might be involved in seed development have not been identified. 

In the present study, we carried out a transcriptomic analysis of common buckwheat seed at three developmental stages (pre-filling stage (S1), filling stage (S2), and initial maturity stage (S3)) with the purpose of investigating its transcriptional dynamics and identifying candidate genes involved in seed development. In addition, some vital seed-development candidate genes were verified by quantitative real-time PCR (qRT-PCR). The results obtained in this study will enhance our understanding of the molecular mechanisms of common buckwheat seed development and provide candidate genes for future common buckwheat seed improvement.

## 2. Results

### 2.1. Overview of Sequencing Data Analysis

To gain insight into the transcriptional dynamics of seed development in common buckwheat, nine libraries from three samples (three biological replicates for each sample) were constructed and subjected to high-throughput RNA-Seq. An overview of the sequencing data is shown in Table 1. A total of 248.53 million raw reads, with an average of 27.61 million raw reads for each library, were yielded. After removing the adaptor sequences and low-quality sequence reads, 242.86 million clean reads were obtained, with an average of 26.98 million clean reads for each library. The Q30 percentages for all libraries exceeded 93%, and the range of GC (Guanine and Cytosine) content for each library was 47.65 to 50.58%. A total of 63.32 to 69.46% of clean reads were mapped to the reference genome for each library. The Pearson’s rank correlation between any two of the three replicates for each sample was >0.95, with 0.979, 0.985, and 0.972 for S1, 0.952, 0.964, and 0.968 for S2, and 0.983, 0.993, and 0.998 for S3 (Appendix A), indicating the high reliability and reproducibility of the data.

### 2.2. Identification and Annotation of Differentially Expressed Genes (DEGs)

To identify the DEGs during the common buckwheat seed development, the gene expression profile clustering was subjected to first analysis. All 54,582 genes were assigned to seven different clusters by the K-means method, which led to the identification of genes with similar expression during all stages of seed development, and indicated that a large number of genes were differentially expressed between different samples (Figure 1). 

Then, gene expression levels among the three developmental stages were subjected to a pairwise comparison. A total of 4619 DEGs were identified, of which 745 (492 up- and 253 down-regulated), 2449 (1506 up- and 943 down- regulated), and 3680 (2162 up- and 1518 down-regulated) were found between S1 and S2, S2 and S3, and S1 and S3, respectively (Figure 2B). Of these DEGs, 304, 576, and 1572 DEGs specifically existed in the S1 vs. S2, S2 vs. S3, and S1 vs. S3 comparisons, respectively; 147, 382, and 1814 DEGs existed both in S1 vs. S2 and S2 vs. S3, S1 vs. S2 and S1 vs. S3, and S2 vs. S3 and S1 vs. S3, respectively; and 88 DEGs were contained in all three comparisons (Figure 2A). 

To further explore the function of these DEGs, Gene Ontology (GO) enrichment analysis was carried out. As a result, 395 (S1 vs. S2), 1244 (S2 vs. S3), and 1878 (S1 vs. S3) DEGs were divided into biological process, cellular component, and molecular function categories (Figure 3 and Appendix A). As shown in Figure 3, the biological process category contained 20 functional groups, in which ‘metabolic process’, ‘cellular process’, and ‘single-organism process’ were the top three represented terms. In the cellular component category, 16 functional groups were enriched, and the most abundant terms were ‘cell part’, ‘cell’, and ‘organelle’. For the molecular function category, 16 functional groups were identified, and the terms with greatest abundance were ‘catalytic activity’, ‘binding’, and ‘transporter activity’. 

To investigate the metabolic pathways involved by the DEGs, Kyoto Encyclopedia of Genes and Genomes (KEGG) analysis was performed. A total of 237 (S1 vs. S2), 789 (S2 vs. S3), and 1244 (S1 vs. S3) DEGs were assigned to 69, 99, and 113 KEGG pathways, respectively (Appendix A). All of the pathways from these three comparison groups could be further divided into five categories: Cellular processing, environmental information processing, genetic information processing, and metabolism and organismal systems (Figure 4 and Appendix A). Among the five groups, the metabolism group had the richest number of pathways. Furthermore, in the S1 vs. S2 comparison, the protein processing in endoplasmic reticulum (ko04141, 25genes), starch and sucrose metabolism (ko00500, 19 genes), amino sugar and nucleotide sugar metabolism (ko00520, 10 genes), and phenylpropanoid biosynthesis (ko00940, 10 genes) were the most abundant pathways (Figure 3). In S2 vs. S3, DEGs were most highly enriched in the ribosome (ko03010, 80 genes), protein processing in endoplasmic reticulum (ko04141, 62 genes), spliceosome (ko03040, 37 genes), and starch and sucrose metabolism (ko00500, 31genes) (Appendix A). The top four DEGs from S1 vs. S3 were mapped to the ribosome (ko03010, 60 genes), starch and sucrose metabolism (ko00500, 57 genes), protein processing in endoplasmic reticulum (ko04141, 53 genes), and carbon metabolism (ko01200, 43 genes) (Appendix A).

### 2.3. Key Genes Involved in the Seed Development of Common Buckwheat

#### 2.3.1. DEGs Involved in Ca^2+^ Signal Transduction Pathways

Genes involved in Ca^2+^ signaling pathway regulate many plant development processes, including seed development [29,30]. According to the GO annotations, 22 DEGs involved in the Ca^2+^ signaling pathway were identified (Table 2). These genes encode calmodulin-like proteins (CMLs), calmodulin binding-like proteins (CBLs), cation/Ca^2+^ exchangers (CCXs), Ca^2+^-dependent protein kinases (CDPKs), CBL-interacting protein kinases (CIPKs), Ca^2+^-transporting ATPase, and Cation/H^+^ antiporter (CHX). As shown in Table 2, these genes display five different expression patterns: Up-regulated both at S2 and S3 (1 *CML*), up-regulated at S2 and down-regulated at S3 (1 *CBL,* 2 *CDPKs,* 1 *CIPK,* and 2 *Ca^2+^-transporting ATPases*), only down-regulated at S2 (1 *CDPK*), only up-regulated at S3 (2 *CMLs*), and only down-regulated at S3 (1 *CBL,* 2 *CCXs*, 1 *CDPK,* 4 *CIPKs*, 2 *Ca^2+^-transporting ATPases,* and 2 *CHXs*).

#### 2.3.2. DEGs Involved in Hormone Signal Transduction Pathways

Hormone signals play an important role in plant seed development. Based on the KEGG enrichment analysis results, 30 DEGs were assigned to “plant hormone signal transduction” (ko04075), including the auxin, CK, GA, BR, ABA, ET, JA, and SA signaling pathways (Table 3). In the auxin signaling pathway, one gene encoding auxin-responsive protein (*IAA*) was up-regulated at S2, five *IAA* genes were up-regulated at S3, and one ortholog of *AFR* encoding for auxin response factor was up-regulated both at S2 and S3 (Table 3). Moreover, one ortholog of *AUX1* encoding for auxin influx carrier and two *IAA* were down-regulated at S3. In the cytokinin pathway, one *ARR* (two-component response regulator) and one *AHP* (histidine-containing phosphotransfer protein) were increased at S2 and S3, respectively (Table 3). In the GA pathway, one *PIF3 (*phytochrome-interacting factor 3) showed up-regulated trends during seed development, and one ortholog of *bHLH127* was increased only at S3 (Table 3). In the ABA pathway, there were two up- and one down-regulated transcripts at S2, which belonged to *PP2C* (protein phosphatase 2C) and *ABF* (encoding for ABA responsive element binding factor), respectively. In addition, one *PYR/PYL* (abscisic acid receptor), one *PP2C,* and one *SnRK2* (serine/threonine protein kinase) were down-regulated at S3 (Table 3). In the ET pathway, two orthologs of *EIN3* (Ethylene-Insensitive Protein 3) were up-regulated at S2 and S3, respectively. Furthermore, two *ETR* (Ethylene Receptor), one *EIN4* (Ethylene-Insensitive Protein 4), and one *CTR1* (serine/threonine-protein kinase) were down-regulated at S3 (Table 3). In the BR pathways, one *BRI1* (Brassinosteroid Insensitive 1) was down-regulated at S3, and other *BRI1* displayed up-regulated at S2 and down-regulated at S3 (Table 3). In addition, one DEG was involved in both the JA and SA pathways, which was up-regulated (*JAZ*) at S3 and down-regulated (*TGA*) at S2 (Table 3), respectively. 

#### 2.3.3. DEGs Involved in TFs

TFs are the key regulatory proteins in seed development. Therefore, the expression dynamics of TF genes in the development of common buckwheat were investigated. According to the RNA-seq data, a total of 2453 TFs were identified as expressed in at least one development phase (Appendix A). The top ten largest TF families were FAR1 (231), C2H2 (185), AP2 (169), bHLH (158), MYB (154), MYB-like (137), ZF-HD (110), NAC (107), bZIP (105), and B3 (97). The comparison among these three development samples found that 41 (30 up- and 11 down-regulated), 82 (47 up- and 35 down-regulated), and 144 (86 up- and 58 down-regulated) TFs had significantly differential expression in S1 vs. S2, S2 vs. S3, and S1 vs. S3, respectively (Appendix A). Among these, the TF families with the largest numbers were *AP2* (9), *bHLH* (11), and *AP2* (22) for these three comparison groups, respectively (Appendix A).

#### 2.3.4. DEGs Involved in Seed Size

Seed size is a key factor to determine seed yield in crops. To identify the genes involved in the seed size of common buckwheat, 95 known seed size-related genes from different plant species were used to homologously search the DEGs (Appendix A). As a result, 17 DEGs were identified as the orthologs of 15 known seed size-related genes (*AtANT*, *AtAP2*, *AtDA1*, *AtDET2*, *AtDWF4*, *AtEOD1*, *AtIKU2*, *AtTTG2*, *OsSLG*, *OsGRF4*, *OsGIF1*, *OsSRS5*, *OsWRKY53*, *OsCYP78A13*, and *OsCYP724B1*) (Appendix A). Of these, two genes (*DWF4* and *CYP724B1*) and one gene (*GIF1*) showed up-regulation only at S2 or at S3, respectively; five genes (*DET2*, *GRF4*, *WRKY53*, *IKU2*, and *TTG2*) were up-regulated during seed development stages; four genes (*SSR5*, *SSR5*, *IKU2*, and *SLG*), with high expression at S1 and S2, were down-regulated at S3; two genes (*CYP78A13* and *ANT*) were up-regulated at S1 compared to S2, and followed by down-regulation at S3; however, three genes (*AP2*, *DA1*, and *EOD1*) displayed the opposite expression pattern with *CYP78A13* and *ANT*, which were down-regulated at S1 compared to S2, and followed by up-regulation at S3 (Figure 5 and Appendix A). 

#### 2.3.5. DEGs Involved in Starch Biosynthesis

To provide insight into the transcriptional dynamics of starch biosynthesis genes, a total of 25 sucrose synthase (*SUS*), 3 UDP glucose pyrophosphorylase (*UGPase*), 14 ADP glucose pyrophosphorylase (*AGPase*), 8 granule bound starch synthase (*GBSS*), 24 starch synthase (*SS*), 4 starch-branching enzyme (*SBE*), and 3 debranching enzyme (*DBE*) genes were identified by using the starch biosynthesis-related genes in *Arabidopsis* to perform a homologous comparative search in the common buckwheat genome database (Appendix A). Of these, four *SUS*, one *UGPase*, and one SBE genes showed up-regulation only at S2; one *SUS*, one *AGPase*, one *GBSS,* and one *SS* displayed down-regulation only at S3; two *SUSs* and one *SS*, and one *SBE* gene were up-regulated at S2, and followed by down-regulation at S3; one DBE gene was down-regulated with all seed development stages (Table 4 and Figure 6).

### 2.4. qRT-PCR Validation of RNA-Seq Results

Based on the results mentioned above, 15 genes which had significantly differential expression during seed development were selected by performing qRT-PCR assays to validate the correlation against RNA-Seq. The results showed that 14 genes had high correlation between the qPCR and RNA-Seq data sets (Figure 7), which thus verified the transcriptomic data.

## 3. Discussion

The seed development of crop plants affects not only seed fate, but also final seed yield and quality. Thus, an in-depth understanding of the molecular mechanisms of seed development is crucial in improving seed yield and quality of crop plants. To date, many genes that contribute to seed development have been identified and functionally verified, and molecular pathways have been deciphered in the model plants *Arabidopsis* and rice [4,5,6,7,8,9]. Furthermore, in some other crop plants, transcriptome analysis has been performed to give insight into the transcriptional dynamics of seed development and provided some clues as to the development of seeds [10,11,12,13,14,15,16,17,18,19,20,21]. In this study, we performed a transcriptome analysis during common buckwheat seed development to give insight into its transcriptional dynamics and find candidate genes that might be involved in seed development.

In total, 242.86 million clean reads were obtained from nine sequencing libraries, and 63.32 to 69.46% clean reads for each library were mapped to the reference genome. The map rate was relatively low, which might have been caused by the draft assembly of the common buckwheat genome [22]. The Q30 percentage was over 93% for each library, and the range of GC content was 47.77–50.58% for each library. Furthermore, the reproducibility was very well (*r* > 0.95) for the three replicates of each sample. These results indicated that the RNA-Seq had higher quality. Through comparison of gene expression patterns between different samples, 4619 DEGs were identified, and the DEGs involved in the Ca^2+^ signaling pathway, hormone signaling pathway, TFs, seed size, and starch biosynthesis were the main focus of this study. 

The Ca^2+^ signal transduction pathways participate in many developmental processes in plants, including pollen tube growth, stem elongation, vascular development, embryogenesis, and seed germination [29,30]. In addition, some studies have also indicated that the Ca^2+^ signal transduction pathways play an important role in seed development with a focus on *CDPK*. For example, several *CDPK* genes were differential or special expression during seed development in rice, maize, and peanut, some of which expressed dominant in the early stage, while others expressed prominently in the latter stage [29,31,32,33,34,35,36], suggesting that different *CDPK* genes might be involved in different development process of seeds. Furthermore, *OsCDPK2*, *OsCDPK11*, and *OsCDPK23* in rice and *RcCDPK1* in castor have been functionally verified to be essential for seed development [32,34,35,36]. In this study, four *CDPK* genes were identified as DEGs during common buckwheat seed development. Among them, two were up-regulated at S2 and down-regulated at S3, and one was only decreased at S3, indicating that they might have an important role in the early stages of common buckwheat seed development. In contrast, the other one was down-regulated at S2, suggesting that it might have an opposed function during common buckwheat seed development with the previous three *CDPK* genes. In addition, we also identified another 18 DEGs involved in the Ca^2+^ signaling pathway, including 3 *CML*, 2 *CBL*, 2 *CCX*, 5 *CIPK*, 4 *Ca^2+^-ATPase*, and 2 *CHX* genes, which displayed similar expression patterns with *CDPK* genes. These results indicate that the Ca^2+^ signaling pathway is conserved in regulatory seed development and that these identified DEGs could be key candidate genes for the seed development of common buckwheat. 

It is well known that hormone signaling, including auxin, CK, GA, BR, ABA, and ET signaling, plays a crucial role in seed development [8,9]. In tartary buckwheat (*Fagopyrum Tararicum*), a dozen of these hormone-related DEGs were identified by RNA-seq, which displayed different expression patterns during seed development [17,18]. In this study, based on KEGG pathway enrichment analysis, we also identified a dozen genes expressed differentially in these hormone signaling pathways during common buckwheat seed development, and also found that some of them have the similar expression patterns with those in tartary buckwheat. Both our study and previous studies have suggested that these hormone signaling pathways are conserved for seed development in different seed plants. Notably, we found one *JAZ* and one *TAG* gene, which are the key genes in the JA and SA signaling pathways, respectively, were also differentially expressed during common buckwheat seed development, implying that the JA and SA signaling pathways probably also have important roles in common buckwheat seed development. 

Seed size is a key factor to determine yield in crops. The identification of genes associated with seed size will support yield improvement. To date, over 90 genes have been identified and functionally validated to regulate seed size in different seed plants [9]. In our study, 17 DEGs were identified as the orthologs of 15 known seed size genes from *Arabidopsis* and rice. Among these, the expression level of the orthologs of *AtAP2*, *AtDA1*, *AtEOD1*, and *OsSLG* were down-regulated during seed development. Interestingly, *AtAP2* [37,38], *AtDA1* [39], *AtEOD1*[39,40], and *OsSLG* [41] have been demonstrated to negatively regulate the seed size in *Arabidopsis* and rice, respectively. This indicates that these genes from common buckwheat may also negatively regulate the seed size. In contrast, the orthologs of several positively-regulated genes for seed size, such as *AtIKU2* [42], *OsGRF4* [43,44], *OsGIF1* [43,44], *OsCYP78A13* [45,46], *OsCYP724B1* [47], *AtDET2* [48], *AtDWF4* [48], *OsWRKY53* [49], *AtANT* [50], and *AtTTG2* [51], were increased at express level in at least one development stage, suggesting that they have a conservative function of positively regulating the seed size of common buckwheat. Notably, the expression level of the orthologs of *OsCYP78A13* and *AtANT* were up-regulated at filling stage and down-regulated at initial maturity stage, indicating that they might regulate the accumulation of seed storage materials and ultimately affect seed size. In addition, an ortholog of *AtIKU2* and two orthologs of *OsSRS5* (positively regulated seed size) were down-regulated at both filling stage and initial maturity stage, suggesting that they might preform function for seed size at the embryogenesis phase, mainly related to cell division and expansion. 

The seeds of common buckwheat contain abundant starch, accounting for over 70% of seed dry weight [23,24,25]. However, the mechanism of starch biosynthesis in common buckwheat remains unclear. It is well known that seven class genes, including *SUS*, *UGPase*, *AGPase*, *GBSS*, *SS*, *SBE*, and *DBE,* catalyze starch biosynthesis [52]. In our data, we identified 25 *SUS*, 3 *UGPase*, 14 *AGPase*, 8 *GBSS*, 24 *SS*, 4 *SBE*, and 3 *DBE* genes. Of these, seven *SUS*, one *AGPase*, one *UGPase*, one *GBSS*, two *SS*, two *SBE*, and one *DBE* genes were differentially expressed during the seed development of common buckwheat. This finding suggests that these identified starch biosynthesis DEGs might be the major functional genes for starch biosynthesis in common buckwheat seed, and could accelerate further functional analysis of them as well. 

Seed development is predominantly regulated by genetic factors and is also greatly influenced by physiological pathways [1,2]. Usually, seed development is involved in seed size change and various nutrients accumulation, which determine crop yield and quality [2]. In this study, we provided insight into the transcriptional dynamics of seed development in common buckwheat at three different stages, and identified DEGs involved in the development of common buckwheat seed. Of these DEGs, some were prominently expressed in the early stage, and some other were highly expressed in the latter stage, indicating different DEGs play different roles during common buckwheat seed development. More importantly, of these DEGs, some genes were identified as the key candidate genes that might participate in the seed development of common buckwheat, including Ca^2+^ signaling, hormone signaling, TFs, seed size, and starch biosynthesis-related genes, and its differential expression patterns during common buckwheat seed development were verified by qRT-PCR. The results are helpful to further dissect the molecular mechanism of buckwheat seed development and provide a basis for buckwheat seed size and quality improvement.

## 4. Materials and Methods 

### 4.1. Plant Material and Sample Collection

The common buckwheat cultivar “Chitian No. 1” was used in this study. It was planted in the growth chamber of the Research Center of Buckwheat Industry Technology, Guizhou Normal University (Lat. 26˚62’ N, 106˚72’ E, Alt. 1168 m), China, in Spring 2018. All plants were grown under natural daylight and subjected to normal management during the growth periods. Seeds were harvested on the 8th, 14th, and 21st days after full bloom, which corresponded to the pre-filling stage (S1), filling stage (S2), and initial maturity stage (S3). For each stage, approximately 200 mixed seeds were collected from the same five plants, and three independent biological replicates were performed. All samples were immediately frozen in liquid nitrogen and stored at −80 °C for RNA-Seq and qRT-PCR validation analyses.

### 4.2. RNA Extraction, Library Construction, and Sequencing

Total RNA was extracted using the EASYspin Plus Plant RNA Kit (Aidlab, Beijing, China) with three replicates for each sample, according to the manufacturer’s instructions. Contaminated DNA in the total RNA samples was removed using DNase I (TAKARA, Dalian, China). The quality and concentration of the total RNA were determined using 1.2% agarose gel electrophoresis and a NanoDrop 2000c spectrophotometer (NanoDrop, Wilmington, DE, USA). Then, the cDNA libraries for Illumina sequencing were constructed using the NEBNext^®^Ultra™ II RNA Library Prep Kit for Illumina^®^ (New England BioLabs, Ipswich, MA, USA). Finally, the constructed cDNA libraries were sequenced using the Illumina XtenPE150 system by Biomarker Technologies Co., Ltd. (Beijing, China).

### 4.3. Analysis of RNA-Seq Data

The raw reads were processed to produce clean reads by removing the adaptor sequences, low-quality sequence reads (*Q* < 20), and poly-N stretches (>10%). Then, the clean reads were mapped to the common buckwheat reference genome (ftp://ftp.kazusa.or.jp/pub/buckwheat/) to obtain the unigenes by using Tophat2 software [53]. Moreover, TopHat alignment using Cufflinks reference annotation was applied to identify the novel transcripts [54]. All unigenes and novel transcripts were functionally annotated by searching against public databases, including NR [55], Swiss-Prot [56], GO [57], COG [58], KOG [59], Pfam [60], and KEGG [61]. 

### 4.4. Identification of DEGs

The transcript abundance of each unigene was calculated and normalized to fragments per kilobase of transcript per killion fragments mapped (FPKM) [62]. Significantly differential gene expression among the three samples was evaluated using the DEseq package [63]. Genes with a threshold of FDR (false discovery rate) value <0.05 and |log2(fold change)| ≥ 1 were assigned as differentially expressed genes (DEGs). The identified DEGs were further subjected to analysis through Gene Ontology (GO) functions, COG (Cluster of Orthologous Groups of proteins), and KEGG pathway analysis. 

### 4.5. Validation of DEGs by qRT-PCR

qRT-PCR analysis was performed to verify the RNA-Seq data. These genes that reflected some of the functional categories and different expression levels were selected. The reference cDNA sequences of these genes were obtained from the common buckwheat genome sequence database. The RT-qPCR primers were designed using Primer 5.0 software according to the reference cDNA sequences (Appendix A). *FeActin7* was used as the internal control. RT-qPCR was conducted using the SYBR premix Ex Taq kit (TaKaRa, Dalian, China) on ABI 7500 Fast Real-time PCR system (ThermoFisher Scientific, Waltham, MA, USA). Each sample was analyzed in triplicate. The relative expression change of each gene was calculated using the 2^−ΔΔ*C*t^ method [64]. The Pearson correlation coefficients between the fold change among different stages from qRT-PCR and from RNA-Seq were calculated using Origin 8.0.

## Figures and Tables

**Figure 1 ijms-20-04303-f001:**
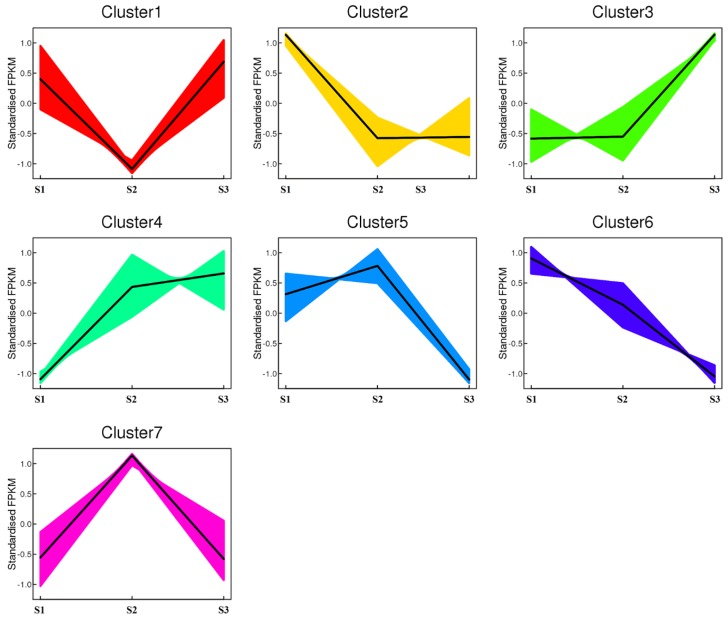
Clustered gene expression profiles from developing tartary buckwheat seeds. The clusters were defined on the basis of genes’ temporal expression profiles in R using K-means. The Y-axis represents the standardized FPKM value of genes, and the X-axis represents the different sample. S1, S2, and S3 stand for the seed samples at pre-filling stage, filling stage, and initial maturity stage, respectively.

**Figure 2 ijms-20-04303-f002:**
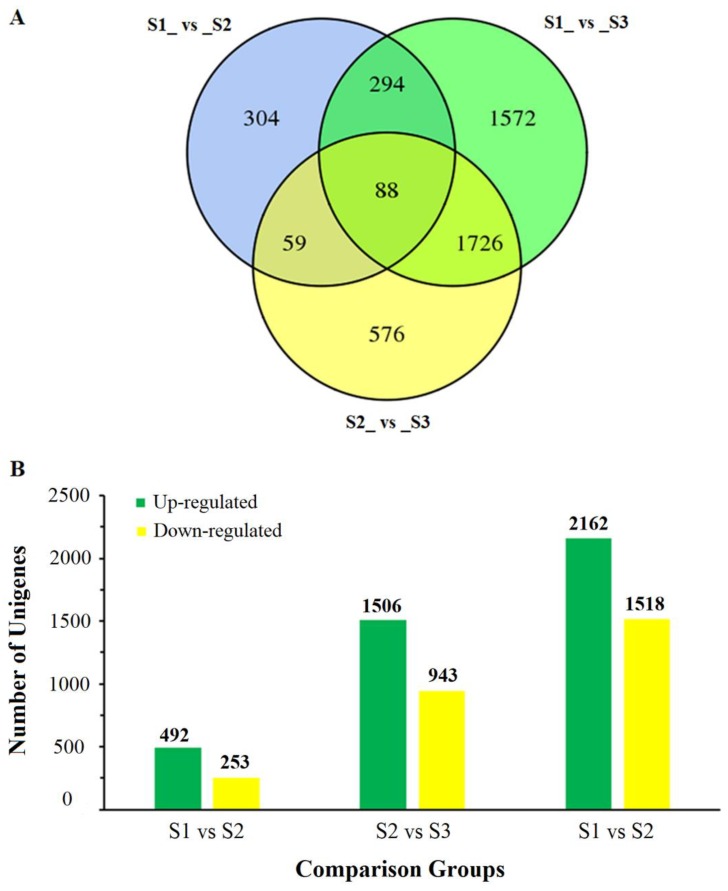
Numbers of specific differentially expressed genes (DEGs) in different comparison groups during common buckwheat seed development. (**A**) Venn diagram for DEGs at different development stages of common buckwheat. (**B**) Numbers of up- and down-regulated DEGs in different comparison groups. S1, S2, and S3 stand for the seed samples at pre-filling stage, filling stage, and initial maturity stage, respectively.

**Figure 3 ijms-20-04303-f003:**
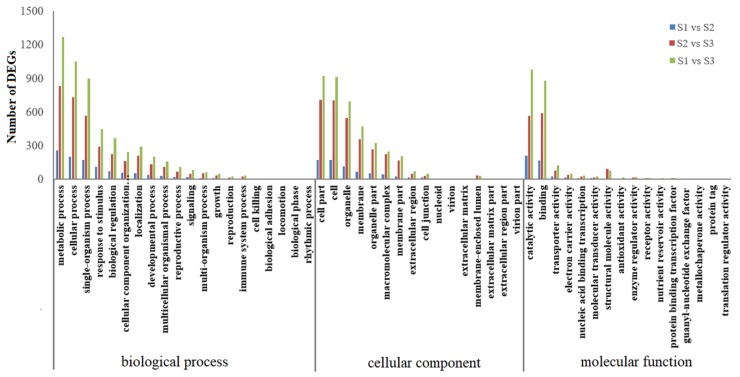
GO enrichment of identified DEGs in developing common buckwheat seeds. S3. S1, S2, and S3 stand for the seed samples at pre-filling stage, filling stage, and initial maturity stage, respectively.

**Figure 4 ijms-20-04303-f004:**
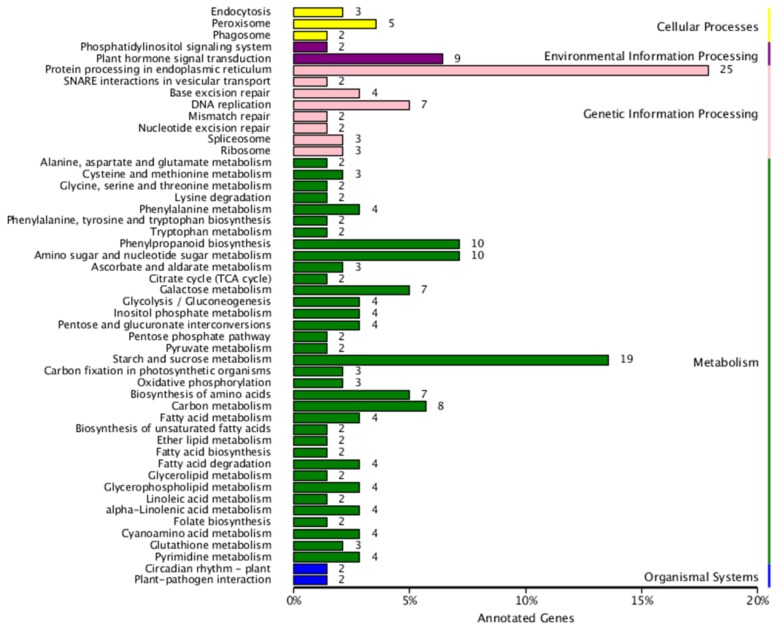
KEGG pathway assignment of the DEGs in S1 vs. S2 comparison group (top 50 pathways according to enrichment factor). S1 and S2 stand for the seed samples at pre-filling stage and filling stage, respectively.

**Figure 5 ijms-20-04303-f005:**
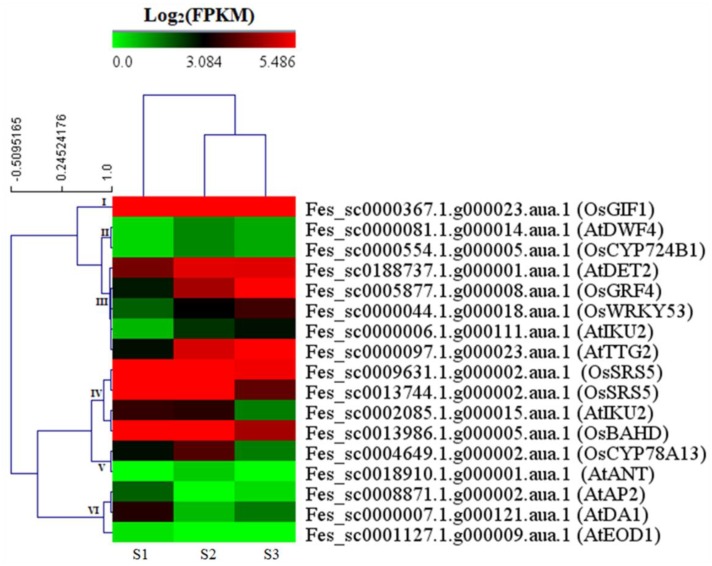
Heat map diagrams of relative expression levels of the differentially expressed orthologs of seed size-related genes in the different development stages of common buckwheat. The heat map was constructed using MeV 4.9.0 based on the Log_2_(FPKM) value of genes. S1, S2, and S3 stand for the seed samples at pre-filling stage, filling stage, and initial maturity stage, respectively.

**Figure 6 ijms-20-04303-f006:**
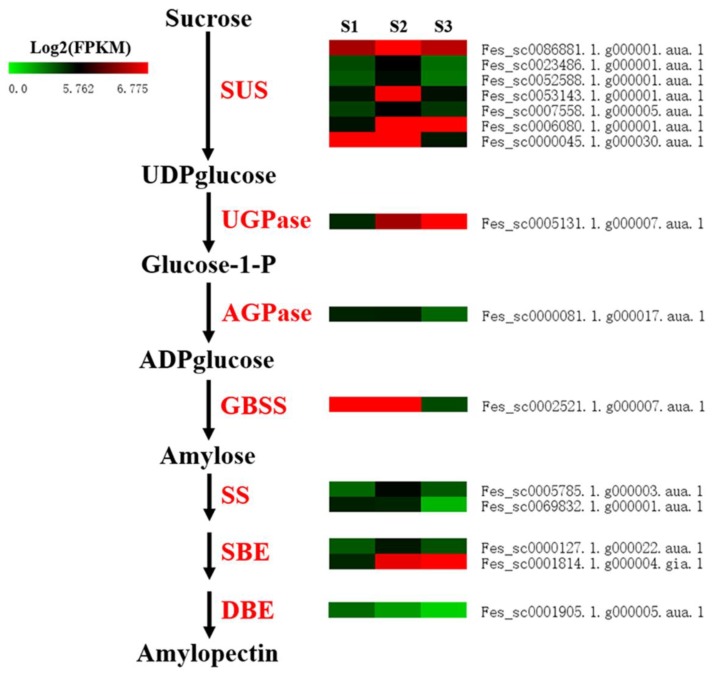
Expression patterns of starch biosynthesis genes that were differentially expressed during seed development. *SUS*: Sucrose synthase, *UGPase*: UDP glucose pyrophosphorylase, *AGPase*: ADP glucose pyrophosphorylase, *GBSS*: Granule bound starch synthase, *SS*: Starch synthase, SBE: Starch-branching enzyme (*SBE*), DBE: Debranching enzyme. S1, S2, and S3 stand for the seed samples at pre-filling stage, filling stage, and initial maturity stage, respectively.

**Figure 7 ijms-20-04303-f007:**
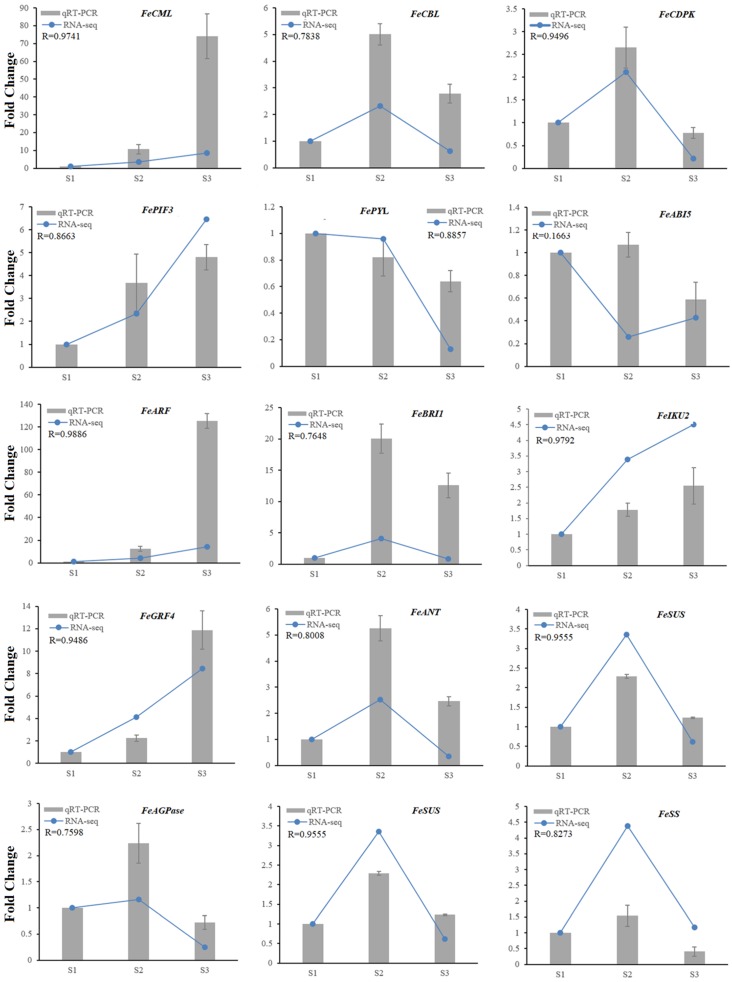
qRT-PCR confirmed 15 DEGs in developing common buckwheat seed. Blue lines represent the RNA-seq values of these selected genes, and grey columns indicate the qRT-PCR results. Relative expression levels of qRT-PCR were calculated using *β-actin* as a standard. Pearson correlation coefficients were calculated by comparing qRT-PCR and RNA-Seq data for each gene across all samples.

**Table 1 ijms-20-04303-t001:** Overview of the RNA-Seq results in developing tartary buckwheat seed.

Item	S1-1	S1-2	S1-3	S2-1	S2-2	S2-3	S3-1	S3-2	S3-3
Raw Reads	25,672,949	22,929,589	24,865,374	25,424,457	27,829,981	30,527,439	25,646,270	37,234,140	28,939,774
Clean Reads	25,494,672	22,747,870	24,566,718	25,250,033	27,625,287	30,212,779	21,406,245	36,901,842	28,656,479
GC Content (%)	47.77	48.46	47.65	48.94	48.82	49.71	48.61	50.58	48.33
Q_30 (%)	93.82	93.65	93.38	93.59	93.80	94.02	94.46	94.18	93.63
Mapped Reads	17,576,027 (68.94%)	14,624,606 (64.29%)	16,309,844 (66.39%)	17,149,822 (67.92%)	17,492,332 (63.32%)	19,840,732 (65.67%)	14,868,778 (69.46%)	25,263,001 (68.46%)	19,801,627 (69.21%)
Unique Mapped Reads	17,327,070 (68.14%)	14,438,073 (63.47%)	16,088,744 (65.49%)	16,942,772 (67.10%)	17,265,804 (62.50%)	19,611,115 (64.91%)	14,671,840 (68.54%)	25,000,998 (67.75%)	19,580,972 (68.33%)
Multiple Mapped Reads	203,957 (0.80%)	186,533 (0.82%)	221,100 (0.90%)	207,050 (0.82%)	226,528 (0.82%)	229,617 (0.76%)	196,938 (0.92%)	262,003 (0.71%)	220,655 (0.88%)

Notes: S1, S2, and S3 stand for the seed samples at pre-filling stage, filling stage, and initial maturity stage, and -1, -2, and -3 represent the three replicates per sample. GC Content (%) represents the percentage of Guanine and Cytosine in clean reads. Q_30 (%) represents the percentage of nucleotides with quality value ≥30.

**Table 2 ijms-20-04303-t002:** DEGs involved in Ca^2+^ signaling pathways. S1, S2, and S3 stand for the seed samples at pre-filling stage, filling stage, and initial maturity stage, respectively. FDR stands for false discovery rate. Log2FC stand for Log2 fold change.

	Gene ID	S1 vs. S2			S2 vs. S3			Annotation
		FDR	Log2FC	up/down	FDR	Log2FC	up/down	
CaM/CML	Fes_sc0060020.1.g000001.aua.1	4.52 × 10^−6^	1.84	up	0.00005	1.23	up	Calcium-like protein CML38
	Fes_sc0006049.1.g000004.aua.1	-	-	-	1.17 × 10^−9^	2.25	up	Calmodulin-like protein 4
	Fes_sc0002338.1.g000002.aua.1	-	-	-	3.71 × 10^−14^	1.74	up	Calmodulin-like protein 9
CBL	Fes_sc0011071.1.g000006.aua.1	7.98 × 10^−14^	1.21	up	0.00021	−1.89	down	Calmodulin binding protein-like
	Fes_sc0011071.1.g000006.aua.1	-	-	-	3.54 × 10^−18^	−1.89	down	Calmodulin binding protein-like
CCX	Fes_sc0000003.1.g000052.aua.1	-	-	-	3.39 × 10^−7^	−1.94	down	Cation calcium exchanger 4
	Fes_sc0000035.1.g000047.aua.1	-	-	-	5.08 × 10^−9^	−2.11	down	Cation calcium exchanger 3
CDPK	Fes_sc0006858.1.g000001.aua.1	0.00016	−3.92	down	-	-	-	Calcium-dependent protein kinase 1
	Fes_sc0219194.1.g000001.aua.1	0.00049	1.56	up	0.00023	−3.15	down	Calcium-dependent protein kinase 8
	Fes_sc0000035.1.g000053.aua.1	6.85 × 10^−11^	1.08	up	0.00013	−3.33	down	Calcium-dependent protein kinase 13
	Fes_sc0080717.1.g000001.aua.1	-	-	-	1.83 × 10^−8^	−4.95	down	Calcium-dependent protein kinase 1
CIPK	Fes_sc0008411.1.g000003.aua.1	4.26 × 10^−7^	1.08	up	0.00024	−2.68	down	CBL-interacting -protein kinase
	Fes_sc0220933.1.g000001.aua.1	-	-	-	3.22 × 10^−6^	−3.43	down	CBL-interacting protein kinase 18
	Fes_sc0097817.1.g000001.aua.1	-	-	-	2.67 × 10^−6^	−2.60	down	CBL-interacting protein kinase 2
	Fes_sc0093645.1.g000001.aua.1	-	-	-	3.39 × 10^−9^	−2.31	down	CBL-interacting protein kinase 5
	Fes_sc0000542.1.g000013.aua.1	-	-	-	0.00004	−2.44	down	CBL-interacting protein kinase 7
Ca^2+^-ATPase	Fes_sc0074374.1.g000001.aua.1	1.88 × 10^−11^	1.24	up	0.00002	−5.85	down	Calcium-transporting ATPase 10
	Fes_sc0049771.1.g000001.aua.1	3.53 × 10^−8^	1.74	up	3.30 × 10^−16^	−2.19	down	Calcium-transporting ATPase 8
	Fes_sc0023460.1.g000001.aua.1	-	-	-	0.00009	−2.64	down	Calcium-transporting ATPase 1
	Fes_sc0009288.1.g000009.aua.1	-	-	-	0.00026	−2.00	down	Calcium-transporting ATPase 10
CHX	Fes_sc0150773.1.g000001.aua.1	-	-	-	0.00008	−7.37	down	Cation/H(+) antiporter 17
	Fesculentum_newGene_621	-	-	-	5.17 × 10^−13^	−4.99	down	Cation/H(+) antiporter 15

**Table 3 ijms-20-04303-t003:** DEGs involved in hormone signaling pathways. S1, S2, and S3 stand for the seed samples at pre-filling stage, filling stage, and initial maturity stage, respectively. FDR stands for false discovery rate. Log2FC stand for Log2 fold change.

	Gene ID	S1 vs. S2			S2 vs. S3			Annotation
		FDR	Log2FC	up/down	FDR	Log2FC	up/down	
Auxin	Fes_sc0003131.1.g000008.aua.1	3.57 × 10^−7^	2.13	up	-	-	-	Auxin responsive protein/*IAA*
	Fes_sc0008308.1.g000001.aua.1	-	-	-	0.00023	−1.73	down	Auxin-responsive protein/*IAA12*
	Fes_sc0076315.1.g000001.aua.1	-	-	-	0.00239	−5.90	down	Auxin transporter-like protein/*AUX1*
	Fes_sc0096151.1.g000001.aua.1	-	-	-	4.49 × 10^−14^	−3.12	down	Auxin-responsive protein/*IAA9*
	Fes_sc0032547.1.g000002.aua.1	1.85 × 10^−6^	2.08	up	0.00626	1.76	up	Auxin response factor 7/*ARF7*
	Fes_sc0001323.1.g000012.aua.1	-	-	-	0.00042	4.27	up	Auxin responsive protein/*IAA*
	Fes_sc0006670.1.g000009.aua.1	-	-	-	3.41 × 10^−9^	2.07	up	Auxin-induced protein/*IAA*
	Fes_sc0007310.1.g000002.aua.1	-	-	-	1.36 × 10^−15^	3.66	up	Auxin responsive protein/*IAA*
	Fes_sc0007969.1.g000004.aua.1	-	-	-	1.90 × 10^−8^	2.19	up	Auxin responsive protein/*IAA*
	Fes_sc0010815.1.g000005.aua.1	-	-	-	7.27 × 10^−14^	2.26	up	Auxin responsive protein/*IAA*
Cytokinine	Fes_sc0001025.1.g000012.aua.1	-	-	-	0.00019	1.93	up	Histidine-containing phosphotransfer protein 1/*AHP*
	Fes_sc0040103.1.g000001.aua.1	0.028	3.44	up	-	-	-	Two-component response regulator /*ARR8*
Gibberellin	Fes_sc0005307.1.g000002.aua.1	5.01 × 10^−18^	1.22	up	0.00011	1.47	up	Transcription factor/*PIF3*
	Fes_sc0000054.1.g000047.aua.1	-	-	-	2.56 × 10^−6^	4.92	up	Transcription factor *bHLH127*
Abscisic acid	Fes_sc0011976.1.g000004.aua.1	-	-	-	0.00118	−2.93	down	Abscisic acid receptor/*PYR/PYL*
	Fes_sc0002839.1.g000004.aua.1	2.99 × 10^−15^	2.38	up				Protein phosphatase 2C/*PP2C*
	Fes_sc0011132.1.g000002.aua.1	0.0001	2.17	up				Protein phosphatase 2C/*PP2C*
	Fes_sc0000642.1.g000010.aua.1	-	-	-	4.93 × 10^−10^	−1.72	down	Protein phosphatase 2C/*PP2C*
	Fes_sc0002743.1.g000004.aua.1	-	-	-	2.60 × 10^−7^	−1.83	down	Serine/threonine-protein kinase/*SRK2A*
	Fes_sc0000024.1.g000035.aua.1	0.00035	−1.93	down	-	-	-	ABSCISIC ACID-INSENSITIVE 5 /*ABF*
Ethylene	Fes_sc0005120.1.g000005.aua.1	-	-	-	4.22 × 10^−15^	−3.46	down	Protein EIN4
	Fes_sc0125064.1.g000001.aua.1	-	-	-	4.90 × 10^−10^	−3.45	down	Ethylene receptor 2/*ETR*
	Fes_sc0043049.1.g000001.aua.1	-	-	-	1.91 × 10^−7^	−2.07	down	Ethylene receptor 1/*ETR*
	Fes_sc0027826.1.g000001.aua.1	-	-	-	0.00006	−4.68	down	Serine/threonine-protein kinase/*CTR1*
	Fes_sc0004642.1.g000013.aua.1	6.60 × 10^−9^	3.43	up				Ethylene insensitive 3/*EIN3*
	Fes_sc0006207.1.g000001.aua.1	-	-	-	0.03248	2.01	up	Ethylene insensitive 3-like protein/*EIN3*
Brassinosteroid	Fes_sc0009187.1.g000001.aua.1	2.80 × 10^−12^	2.02	up	0.01759	−2.33	down	BRASSINOSTEROID INSENSITIVE 1/*BRI1*
	Fes_sc0361928.1.g000001.aua.1	-	-	-	3.83 × 10^−7^	−3.60	down	BRASSINOSTEROID INSENSITIVE 1/*BRI1*
Jasmonic acid	Fes_sc0000770.1.g000005.aua.1	-	-	-	0.00011	2.07	up	Protein TIFY 6B
Salicylic acid	Fes_sc0002899.1.g000004.aua.1	0.00027	−1.76	down	-	-	-	bZIP transcription factor

**Table 4 ijms-20-04303-t004:** DEGs involved in starch biosynthesis. S1, S2, and S3 stand for the seed samples at pre-filling stage, filling stage, and initial maturity stage, respectively. FDR stands for false discovery rate. Log2FC stand for Log2 fold change.

	Gene ID	S1 vs. S2			S2 vs. S3			Annotation
		FDR	Log2FC	up/down	FDR	Log2FC	up/down	
SUS	Fes_sc0086881.1.g000001.aua.1	1.06 × 10^−9^	1.74	up	3.08 × 10^−14^	−1.66	down	Sucrose synthase 2
	Fes_sc0023486.1.g000001.aua.1	2.69 × 10^−6^	1.75	up	5.57 × 10^−9^	−2.47	down	Sucrose synthase 3 isoform 4
	Fes_sc0052588.1.g000001.aua.1	0.00015	1.73	up	-	-	-	Sucrose synthase 4
	Fes_sc0053143.1.g000001.aua.1	0.00627	1.57	up	-	-	-	Sucrose synthase 4
	Fes_sc0007558.1.g000005.aua.1	4.33 × 10^−14^	1.51	up	-	-	-	Sucrose synthase 3
	Fes_sc0006080.1.g000001.aua.1	0.00001	1.72	up	-	-	-	Sucrose synthase 1
	Fes_sc0000045.1.g000030.aua.1	-	-	-	0.00035	−1.87	down	Sucrose synthase
UGPase	Fes_sc0005131.1.g000007.aua.1	3.89 × 10^−8^	1.33	up	-	-	-	UDP glucose pyrophosphorylase
AGPase	Fes_sc0000081.1.g000017.aua.1	-	-	-	1.42 × 10^−6^	−2.09	down	ADP-glucose pyrophosphorylase
GBSS	Fes_sc0002521.1.g000007.aua.1	-	-	-	0.00001	−2.75	down	Granule-bound starch synthase 1
SS	Fes_sc0005785.1.g000003.aua.1	2.02 × 10^−11^	2.13	up	2.93 × 10^−10^	−1.89	down	Starch synthase 1
	Fes_sc0069832.1.g000001.aua.1	-	-	-	1.87 × 10^−6^	−3.03	down	Starch synthase 3
SBE	Fes_sc0000127.1.g000022.aua.1	0.00027	1.38	up	1.13 × 10^-12^	−1.32	down	Starch-branching enzyme
	Fes_sc0001814.1.g000004.gia.1	0.00268	1.37	up	-	-	-	Starch-branching enzyme
DBE	Fes_sc0001905.1.g000005.aua.1	3.21 × 10^−12^	−1.53	up	4.66 × 10^−12^	2.21	down	Debranching enzyme

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
