# Peer review of "Transcriptome Analysis Reveals Key Seed-Development Genes in Common Buckwheat (*Fagopyrum esculentum*)"

_ijms, 2019, doi:10.3390/ijms20174303_

Round 1

Reviewer 1 Report

In this study by Li et al, the authors perform an RNAseq analysis on buckweed seeds with the aim of identifying genes potentially involved in seed development. While this can be of potential interest, I do not find the analysis to be sufficiently detailed. It is not clear how the seed stages were determined and selected. Furthermore, whole seeds were analyzed, containing embryo, endosperm and seed coat. This confounds the data and may lead to misrepresentation of differentially expressed genes (DEGs). Overall, the manuscript is quite descriptive and the novelty of the study is not made obvious to the reader. The results section has little detail, which makes it hard to draw significant amounts of information from the analysis.

Some issues that I found were:

- The DEG analysis loses some power due to the pairwise comparison. This approach does not allow identifying genes that are similarly expressed during all stages of seed development. The study would be a lot more powerful with the inclusion of a pre-fertilization sample to the analysis.

It is not clear how old were the seeds collected for the RNAseq analysis. How were the growth stages determined?

- What was the genome coverage of the RNAseq?

- The DEG analysis of Fig 1a is not clear to me. If this is a pairwise comparison, how come are there 88 DEGs commonly deregulated in all samples? What samples are these genes being compared to?

- The letter size in Figures 2 and 3 is too small. It makes it very hard to read the text.

- It is not clear to me what kind of information can be taken from Fig 2. Are these GO groups only the ones that were over-represented in the RNAseq data? This data should be explained in more detail.

- Line 136 - Should provide references for these statements.

- For the qPCR experiments, how does the housekeeping gene behave along seed development?

Author Response

First of all, the authors would like to express their sincere thanks to the anonymous reviewers for helpful comments and suggestions. The explanation of the modifications as well as corrections in this revision can be arranged as follows (comment numbers are in 1:1 correspondence with the reviewers comments).

Comments

In this study by Li et al, the authors perform an RNA-seq analysis on buckweed seeds with the aim of identifying genes potentially involved in seed development. While this can be of potential interest, I do not find the analysis to be sufficiently detailed. It is not clear how the seed stages were determined and selected. Furthermore, whole seeds were analyzed, containing embryo, endosperm and seed coat. This confounds the data and may lead to misrepresentation of differentially expressed genes (DEGs). Overall, the manuscript is quite descriptive and the novelty of the study is not made obvious to the reader. The results section has little detail, which makes it hard to draw significant amounts of information from the analysis.

Some issues that I found were:

Point 1: The DEG analysis loses some power due to the pairwise comparison. This approach does not allow identifying genes that are similarly expressed during all stages of seed development. The study would be a lot more powerful with the inclusion of a pre-fertilization sample to the analysis.

Response 1: Thank you for your professional suggestions. To display the genes that were similarly expressed during all stages of seed development, cluster gene expression profiles were created in R using kmeans. A total of seven unique cluster gene profiles were defined. In each defined cluster, all genes displayed the similar expression pattern during all stages of seed development. Please see Figure 1 and lines 92-96 in the revised manuscript. In addition, thank you for your professional suggestions that providing a negative control data (such as a pre-fertilization sample) which can make our manuscript better. Unfortunately, in this study, we have not the negative control data. But then we will perform such experiments if there are similar studies in the future.

Point 2: It is not clear how old were the seeds collected for the RNA-seq analysis. How were the growth stages determined?

Response 2: For RNA-seq analysis, seeds were collected on the 8th, 14th, and 21st days after full bloom, which corresponding to the pre-filling stage (S1), filling stage (S2) and initial maturity stage (S3). We have added the related information. Please see lines 357-359 in the revised manuscript. The pre-filling stage (S1) and filling stage (S2) were determined according to whether or not existing the milk-white seriflux when pressing the seeds with the fingers. The initial maturity stage (S3) was determined by the color of seed-case which began to turn yellow.

Point 3: What was the genome coverage of the RNA-seq?

Response 3: We are sorry that we did not found the “the genome coverage of the RNA-seq” description in the manuscript and the supplementary materials. Therefore, we can not to explain this question.

Point 4: The DEG analysis of Fig 1a is not clear to me. If this is a pairwise comparison, how come are there 88 DEGs commonly deregulated in all samples? What samples are these genes being compared to?

Response 4: The Fig 1a is the Venn diagram of these three comparison groups (S1 vs. S2, S2 vs. S3, and S1 vs. S3). The intersection of the three comparison groups represents the DEGs that existed in all the three comparison groups, which shown that the expression of these genes at a spectific stage changed greatly relative to any other stage. The samples were S1, S2, and S3 for these genes being compared.

Point 5: The letter size in Figures 2 and 3 is too small. It makes it very hard to read the text.

Response 5: We have modified these two figures according to your suggestion, and the letters in these two figures became clearer. Please see Figure 3 and Figure 4 in the in the revised manuscript.

Point 6: It is not clear to me what kind of information can be taken from Fig 2. Are these GO groups only the ones that were over-represented in the RNA-seq data? This data should be explained in more detail.

Response 6:  we have modified this Fig 2 to Fig 3 in the revised manuscript. These GO groups represent the all GO categories of all DEGs from these three comparison groups.

Point 7: Line 136 - Should provide references for these statements.

Response 7: We have provide the relative references. Please see line 157 and references 29-30 in the revised manuscript.

Point 8: For the qPCR experiments, how does the housekeeping gene behave along seed development?

Response 8: In previous study, we selected 18 reference genes (including FeActin7: Fes_sc0000626.1.g000008.aua.1) to evaluate its expression stability in different tissues of common buckwheat, and found that the FeACT7 has good stability in the developing tartary buckwheat seeds. Therefore, it was selected as reference gene in the study.

Reviewer 2 Report

Comments

The manuscript entitled “Transcriptome analysis reveals key seed-development genes in common buckwheat (Fagopyrum esculentum)” by Li et. alseems interesting. However, the data haven’t been discussed justifiably. Authors are requested to be specific on their findings while comparing/relating to findings from earlier studies.

Major comments

Please clarify whether it was technical or sampling replication within each of S1, S2, and S3. Authors identified DEGs based on pairwise comparison among 3-stage samples. It would have been better if authors included negative control data as well, such as non-pollinated and/or seed development mutants (eg des5). The discussion section lacks GO analysis. Additionally, the manuscript has not discussed on up and/or downregulation of particular genes in light of earlier studies. Instead, the differential expression (both up and down-regulation) of the genes were taken equivalent to earlier studies which showed their upregulated status crucial for seed development (eg. CDPK). In case of CDPKs, the data presented in the manuscript shows their down-regulated status in developing seeds. Authors are requested to elaboraton their findings satisfactorily

Author Response

First of all, the authors would like to express their sincere thanks to the anonymous reviewers for helpful comments and suggestions. The explanation of the modifications as well as corrections in this revision can be arranged as follows (comment numbers are in 1:1 correspondence with the reviewers comments).

Point 1: The manuscript entitled “Transcriptome analysis reveals key seed-development genes in common buckwheat (Fagopyrum esculentum)” by Li et al. seems interesting. However, the data haven’t been discussed justifiably. Authors are requested to be specific on their findings while comparing/relating to findings from earlier studies.

Response 1: we have performed this discussion according to your professional suggestions as soon as possible. Please see the discussion section in the revised manuscript.

Point 2: Please clarify whether it was technical or sampling replication within each of S1, S2, and S3.

Response 2: As described lines 359-361, the S1, S2, and S3 stand for the seed samples at pre-filling stage, filling stage, and initial maturity stage. Therefore, it was not technical or sampling replication within each of S1, S2, and S3.

Point 3: Authors identified DEGs based on pairwise comparison among 3-stage samples. It would have been better if authors included negative control data as well, such as non-pollinated and/or seed development mutants (eg des5).

Response 3: Thank you for your professional suggestions, which can make our manuscript becomes better. Unfortunately, in this study, we have not the negative control data. But then, we will perform such experiments if there are similar studies in the future.

Point 4: The discussion section lacks GO analysis.

Response 4: In this study, we main focused on the DEGs involved in the Ca2+ signaling pathway, hormone signaling pathway, TFs, seed size, and starch biosynthesis, which might have important value for the improvement of common buckwheat seed yield and quality. Therefore, we major discussed these DEGs, and didn’t discuss GO analysis.

Point 5: Additionally, the manuscript has not discussed on up and/or downregulation of particular genes in light of earlier studies. Instead, the differential expression (both up and down-regulation) of the genes were taken equivalent to earlier studies which showed their upregulated status crucial for seed development (eg. CDPK). In case of CDPKs, the data presented in the manuscript shows their down-regulated status in developing seeds. Authors are requested to elaboraton their findings satisfactorily.

Response 5: We have corrected these questions as far as possible. Please see lines 286-297 in the revised manuscript.

Reviewer 3 Report

Overall:

The abstract and discussion could include wider context of the work. The authors underplay the potential impact on crop yield and quality; some insight to the next steps would be helpful. 

The figures need to be clearer. They are not easy to read and size of text, graphics and annotation should be increased.

Specific points in each section are highlighted below.

Abstract:

This could be strengthened by highlighting some seed developmental or growth traits of interest for improving yield and quality. This would help create relevance and impact of the work. 

Line 16. Change 'essential and complicated process to determine crop ...' to 'essential and complex process that determine crop ...'

Lines 16-18. State why buckwheat is important. 

Line 21. Add 'gene ontology'. Although there is no need to define 'KEGG analysis' in the abstract, it does need to be explained in line 114.

Introduction:

This is presented very well, with background to the crop, seed and genetic analysis to date. 

Line 38. Delete 'the' from 'the seed'.

Line 42. Delete 'the' from 'the environmental'.

Line 48. Change 'performed in' to 'been undertaken in'.

Line 57. Delete 'obviously'.

Line 58. Replace ''is very important to' with 'is of considerable interest to'.

Results:

Line 75. Change 'To insight into' to 'To gain insight into'.

As the use of different developmental stages is a key part of this study, I suggest that these are mentioned in lines to 83-84. That is, three replicates for each of the three developmental stages.

Table 1. Add more detail to the legend. Include developmental stage and explain the replication (i.e. heading, row 1). Also explain the data in parentheses.  

Line 87. Add 'differentially expressed genes', to DEG. If this is done, then Figure 1 legend (line 97) can refer to 'specific DEGs' rather than 'specific expressed genes (DEGs)'.

Line 88. Delete 'the' from 'the gene expression'.

Line 89. Change 'among these three samples were' to 'among the three developmental stages were'.

Figure 1. Both the bar chart and venn diagram can be made larger. The text in the bar chart is very small. 

Figure 2. This is very difficult to read. Discuss a revised layout with the journal. For example A, B and C could be stacked.

Line 114. Explain KEGG. E.g. add 'Kyoto Encyclopedia of Genes and Genomes', as appropriate.

Figure 3. As comment for Fig. 2.

Table 2. A similar comment to Table 1. Make the legend is more informative by explaining the developmental stages and their comparison. And explain the column headings e.g. FDR.

Table 3. As comment for Table 2. Where tables have common layout, then refer to the previous legend/s.

Line 171. Open the space between section title and Table 3.

Line 171. Add 'transcription factors' to the title.

Figure 4. As for Figs. 2 and 3.

Table 4. As for Tables 2 and 3.

Check spacing between Figure 5 and Table 4.

Figure 6. Could this be presented as five rows of three graphs to improve clarity?

Discussion:

Line 224. Explain what is meant by 'seed fate'. Should the first sentence read as 'The seed development of crop plants affects not only individual seeds, but also final crop yield and quality'?

Lines 225-226. Change 'development is helpful in improving the seed yield ...' to 'development is crucial in improving seed yield ...'.

Line 226. Change '... genes that participated in seed...' to '...genes that contribute to seed...'.

Line 235. Change 'relative lower' to 'relatively low'.

Line 262. Delete 'seed' from 'seed yield'.

Line 263. Change 'will be great help to seed yield improvement' to 'will support yield improvement'. 

Line 280. Change 'of the seeds' to of seed'.    

Lines 281-282. Change 'remains largely unclear' to 'remains unclear'.

Lines 292. Delete 'theoretical' from 'theoretical basis'.

The final paragraph (lines 289-294) in discussion should be expanded by referring back to the original statements about crop yield and quality (in the abstract and introduction). For example, (1) how might differences in gene expression or molecular or physiological pathways have benefits for components of yield and/or quality in buckwheat?  (2) Is there any evidence that genes or gene expression identified this study have had any value in yield or grain quality improvement in other crop species?  (3) Is there any evidence for wider variation in seed traits or development in other cultivars of buckwheat? Such information could help to inform further study. 

Author Response

First of all, the authors would like to express their sincere thanks to the anonymous reviewers for helpful comments and suggestions. The explanation of the modifications as well as corrections in this revision can be arranged as follows (comment numbers are in 1:1 correspondence with the reviewers comments).

Comments

The abstract and discussion could include wider context of the work. The authors underplay the potential impact on crop yield and quality; some insight to the next steps would be helpful. The figures need to be clearer. They are not easy to read and size of text, graphics and annotation should be increased.

Specific points in each section are highlighted below.

Point 1: Abstract:

Point 1.1: This could be strengthened by highlighting some seed developmental or growth traits of interest for improving yield and quality. This would help create relevance and impact of the work.

Response 1.1: We have corrected the question, and highlighted by RED in the revised manuscript. Please see lines 16-17 in the revised manuscript.

Point 1.2: Line 16. Change 'essential and complicated process to determine crop ...' to 'essential and complex process that determine crop ...'

Response 1.2: We have corrected these errors, and highlighted by RED in the revised manuscript. Please see lines 16-17 in the revised manuscript.

Point 1.3: Lines 16-18. State why buckwheat is important.

Response 1.3: We have stated the importance of common buckwheat. Please see lines 17-19 in the revised manuscript.

Point 1.4: Line 21. Add 'gene ontology'. Although there is no need to define 'KEGG analysis' in the abstract, it does need to be explained in line 114.

Response 1.4: We have corrected these questions, and highlighted by RED in the revised manuscript. Please see line 23 and lines 134-135 in the revised manuscript.

Point 2: Introduction:

This is presented very well, with background to the crop, seed and genetic analysis to date.

Point 2.1: Line 38. Delete 'the' from 'the seed'.

Response 2.1: We have corrected the error. Please see line 40 in the revised manuscript.

Point 2.2: Line 42. Delete 'the' from 'the environmental'.

Response 2.2: We have corrected the error. Please see line 44 in the revised manuscript.

Point 2.3: Line 48. Change 'performed in' to 'been undertaken in'.

Response 2.3: We have changed 'performed in' to 'been undertaken in', and highlighted by RED in the revised manuscript. Please see line 50 in the revised manuscript.

Point 2.4: Line 57. Delete 'obviously'.

Response 2.4: We have deleted 'obviously'. Please see line 59 in the revised manuscript.

Point 2.5: Line 58. Replace ''is very important to' with 'is of considerable interest to'.

Response 2.5: We have corrected the question, and highlighted by RED in the revised manuscript. Please see line 60 in the revised manuscript.

Point 3: Results:

Point 3.1: Line 75. Change 'To insight into' to 'To gain insight into'.

Response 3.1: We have changed 'To insight into' to 'To gain insight into', and highlighted by RED in the revised manuscript. Please see line 77 in the revised manuscript.

Point 3.2: As the use of different developmental stages is a key part of this study, I suggest that these are mentioned in lines to 83-84. That is, three replicates for each of the three developmental stages.

Response 3.2: We provided all the correlation coefficients of any two of the three replicates for each stage. Please see lines 86-87 in the revised manuscript.

Point 3.3: Table 1. Add more detail to the legend. Include developmental stage and explain the replication (i.e. heading, row 1). Also explain the data in parentheses. 

Response 3.3: We have added the corresponding content in Table 1 notes. Please see lines 89-90 in the revised manuscript.

Point 3.4: Line 87. Add 'differentially expressed genes', to DEG. If this is done, then Figure 1 legend (line 97) can refer to 'specific DEGs' rather than 'specific expressed genes (DEGs)'.

Response 3.4: Because the 'differentially expressed genes' has been firstly defined as in lines 23 in the revised manuscript, we did not add 'differentially expressed genes', to DEG. However, we have changed the 'specific expressed genes (DEGs)' to 'specific DEGs' in Figure 2 legend in the revised manuscript. Please see line 115 in the revised manuscript.

Point 3.5: Line 88. Delete 'the' from 'the gene expression'.

Response 3.5: We have corrected the error. Please see line 106 in the revised manuscript.

Point 3.6: Line 89. Change 'among these three samples were' to 'among the three developmental stages were'.

Response 3.6: We have changed 'among these three samples were' to 'among the three developmental stages were'. Please see line 106 in the revised manuscript.

Point 3.7: Figure 1. Both the bar chart and venn diagram can be made larger. The text in the bar chart is very small.

Response 3.7: We have corrected the question of the Figure. Please see Figure 2 in the revised manuscript.

Point 3.8: Figure 2. This is very difficult to read. Discuss a revised layout with the journal. For example A, B and C could be stacked.

Response 3.8: We have modified this figure and provided the GO analysis information. Please see Figure 3 and Table S2 in the revised manuscript.

Point 3.9: Line 114. Explain KEGG. E.g. add 'Kyoto Encyclopedia of Genes and Genomes', as appropriate.

Response 3.9: We have corrected the question, and highlighted by RED in the revised manuscript. Please see lines 134-135 in the revised manuscript.

Point 3.10: Figure 3. As comment for Fig. 2.

Response 3.10: We modified this figure. Please see Figure 4 in the in the revised manuscript.

Point 3.11: Table 2. A similar comment to Table 1. Make the legend is more informative by explaining the developmental stages and their comparison. And explain the column headings e.g. FDR.

Response 3.11: We have added the corresponding content in Table 2 notes. Please see lines 167-169 in the revised manuscript.

Point 3.12: Table 3. As comment for Table 2. Where tables have common layout, then refer to the previous legend/s.

Response 3.12: We have added the corresponding content in Table 3 notes. Please see lines 194-196 in the revised manuscript.

Point 3.13: Line 171. Open the space between section title and Table 3.

Response 3.13: we have changed this question. Please see line 197 in the revised manuscript.

Point 3.14: Line 171. Add 'transcription factors' to the title.

Response 3.14: Because the 'transcription factors' has been firstly defined as TFs in lines 26 in the revised manuscript, we did not add 'transcription factors' to the title.

Point 3.15: Figure 4. As for Figs. 2 and 3.

Response 3.15: We modified this figure. Please see Figure 5 in the revised manuscript.

Point 3.16: Table 4. As for Tables 2 and 3.

Response 3.16: We have added the corresponding content in Table 1 notes. Please see lines 239-241 in the revised manuscript.

Point 3.17: Check spacing between Figure 5 and Table 4.

Response 3.17: we have changed the spacing between Figure 5 and Table 4. Please see Figure 6 and Table 4 in the revised manuscript.

Point 3.18: Figure 6. Could this be presented as five rows of three graphs to improve clarity?

Response 3.18: We have modified this Figure according to your professional suggestion. Please see Figure 7 in the revised manuscript.

Point 4: Discussion:

Point 4.1: Line 224. Explain what is meant by 'seed fate'. Should the first sentence read as 'The seed development of crop plants affects not only individual seeds, but also final crop yield and quality'?

Response 4.1: We want to express the means of 'seed fate' is ‘the determines the successful racial continuation of seed plants’ as described in Introduction.

Point 4.2: Lines 225-226. Change 'development is helpful in improving the seed yield ...' to 'development is crucial in improving seed yield ...'

Response 4.2: we have changed 'development is helpful in improving the seed yield ...' to 'development is crucial in improving seed yield ...', and highlighted by RED in the revised manuscript. Please see lines 263-264 in the revised manuscript.

Point 4.3: Line 226. Change '... genes that participated in seed...' to '...genes that contribute to seed...'.

Response 4.3: we have changed '... genes that participated in seed...' to '...genes that contribute to seed...', and highlighted by RED in the revised manuscript. Please see lines 264-265 in the revised manuscript.

Point 4.4: Line 235. Change 'relative lower' to 'relatively low'.

Response 4.4: we have changed 'relative lower' to 'relatively low', and highlighted by RED in the revised manuscript. Please see line 274 in the revised manuscript.

Point 4.5: Line 262. Delete 'seed' from 'seed yield'.

Response 4.5: we have deleted 'seed' from 'seed yield'. Please see line 312 in the revised manuscript.

Point 4.6: Line 263. Change 'will be great help to seed yield improvement' to 'will support yield improvement'.

Response 4.6: we have changed 'will be great help to seed yield improvement' to 'will support yield improvement', and highlighted by RED in the revised manuscript. Please see line 313 in the revised manuscript.

Point 4.7: Line 280. Change 'of the seeds' to ‘of seed'.   

Response 4.7: we have changed 'of the seeds' to ‘of seed', and highlighted by RED in the revised manuscript. Please see line 330 in the revised manuscript.

Point 4.8: Lines 281-282. Change 'remains largely unclear' to 'remains unclear'.

Response 4.8: we have changed 'remains largely unclear' to 'remains unclear', and highlighted by RED in the revised manuscript. Please see lines 331-332 in the revised manuscript.

Point 4.9: Lines 292. Delete 'theoretical' from 'theoretical basis'.

Response 4.9: we have deleted 'theoretical' from 'theoretical basis', and highlighted by RED in the revised manuscript. Please see line 351 in the revised manuscript.

Point 4.10: The final paragraph (lines 289-294) in discussion should be expanded by referring back to the original statements about crop yield and quality (in the abstract and introduction). For example, (1) how might differences in gene expression or molecular or physiological pathways have benefits for components of yield and/or quality in buckwheat?  (2) Is there any evidence that genes or gene expression identified this study have had any value in yield or grain quality improvement in other crop species?  (3) Is there any evidence for wider variation in seed traits or development in other cultivars of buckwheat? Such information could help to inform further study.

Response 4.10: Thank you very much for your professional suggestions. We have corrected these questions as far as possible. Please see lines 339-351 in the revised manuscript.

Round 2

Reviewer 1 Report

The authors addressed my comments in a satisfactory manner. I have no scientific objections to the work being published in IJMS.

Reviewer 2 Report

Good.